

# Genes of the pig, *Sus scrofa*, reconstructed with EvidentialGene

Donald G. Gilbert

Indiana University, Bloomington, IN, USA

## ABSTRACT

The pig is a well-studied model animal of biomedical and agricultural importance. Genes of this species, *Sus scrofa*, are known from experiments and predictions, and collected at the NCBI reference sequence database section. Gene reconstruction from transcribed gene evidence of RNA-seq now can accurately and completely reproduce the biological gene sets of animals and plants. Such a gene set for the pig is reported here, including human orthologs missing from current NCBI and Ensembl reference pig gene sets, additional alternate transcripts, and other improvements. Methodology for accurate and complete gene set reconstruction from RNA is used: the automated SRA2Genes pipeline of EvidentialGene project.

## INTRODUCTION

Precision genomics is essential in medicine, environmental health, sustainable agriculture, and research in biological sciences (*Goldfeder et al., 2016*). Yet the popular genome informatics methods lag behind the high levels of accuracy and completeness in gene construction that is attainable with today's accurate RNA-seq data.

To demonstrate the accuracy and completeness of gene set reconstruction from expressed gene pieces (RNA-seq) alone, excluding chromosome DNA or other species genes, the pig is a good choice. The pig has well-constructed, partly curated gene sets produced by the major genomics centers NCBI and Ensembl, and is one of seven RefSeq top-level model organisms. Gene sets are based on extensive expressed sequences dating from the 1990s. Pig has a well-assembled chromosome set (*Groenen et al., 2012*), improved in 2017, and contributions of experimental gene evidence from many projects of agricultural and biomedical focus. Published RNA-seq from over 2,000 samples puts the pig among the top 10 of model animals and plants. Yet there is just one public transcript assembly from these many pig studies, from blood samples only.

If successful, this demonstration can be used to improve reference genes for this species. It will demonstrate to others how to produce reliably accurate gene sets. Unreliability in gene sets is a continuing problem, measurable from missing and fragment orthologs (*Trachana et al., 2011*; *Tekaia, 2016*; *Simao et al., 2015*; *Waterhouse et al., 2018*). Reasons for unreliability are many, including errors from sequencing and assembly, mis-modeling of complex genes, and errors propagated from public databases. The EvidentialGene

Corresponding author
Donald G. Gilbert,
gilbertd@indiana.edu

project aims for a reliable solution that others can use, simple in concept, to obtain accurate gene sets from a puzzle box full of gene pieces.

Gene sets reconstructed by the author are more accurate by objective measures of homology and expression recovery, than those of the same species produced by popular methods. EvidentialGene has been developed in conjunction with reconstruction of gene sets for popular and model animals and plants such as arabidopsis, corn plant, chocolate tree, zebrafish, atlantic killifish, mosquitos, jewel wasp, and water fleas (*Gilbert, 2012*, *2013*, *2016*, *2017*). These are more accurate than the same species sets produced by NCBI Eukaryotic Genome Annotation Pipeline (EGAP, *Thibaud-Nissen et al., 2013*), Ensembl gene annotation pipeline (*Curwen et al., 2004*), MAKER genome-gene modeling (*Holt & Yandell, 2011*), Trinity RNA assembly (*Grabherr et al., 2011*), Pacific Biosciences long RNA assemblies, and others.

Gene sets reconstructed by others using EvidentialGene methods are also more accurate (*Nakasugi et al., 2014*, *Mamrot et al., 2017*), in independent assessments. However, some investigators do not apply necessary details of EvidentialGene methodology, or modify portions in ways that reduce accuracy. One impetus for this work is engineering a full, automated pipeline that others can more readily use, for these validated methods. Herein is reported a good quality gene set published in databases to aid future research and database improvements. The automated pipeline of SRA2Genes is introduced here with accurate genes for the pig model. EvidentialGene methodology will be detailed in other papers.

## MATERIALS AND METHODS

Materials for this pig gene set reconstruction are primarily four RNA source projects that were selected from over 100 in the SRA database, based on tissue sampling, methodology, and other factors. These four and gene evidence data materials are listed below in "Data and Software Citations." The NCBI SRA web query system allows search and retrieval by species and attributes including tissue, sex, sample type, and instrument type. Queries used to select these materials are listed in Data Citations. Evidence data sets for validation of this pig gene set include reference proteomes of human, mouse, cow and zebrafish, vertebrate conserved single copy gene proteins, and pig chromosome assembly. Data sets for comparison include recent NCBI and Ensembl pig genes, and expressed pig sequences (see Data Citations).

The selected RNA projects are *pig1a* (PRJNA416432, China Agricultural University), *pig2b* (PRJNA353772, Iowa State University, USDA-ARS), *pig3c* (PRJEB8784, Univ. Illinois), and *pig4e* (PRJNA255281, Jiangxi Agricultural University, Nanchang, China). These four include 26 read sets of 1,157,824,292 read pairs, or 106,654 megabases. All these are paired-end reads from recent-model Illumina sequencers, ranging from 75 to 150 bp read length. *Pig3c* includes adult female and male tissues of muscle, liver, spleen, heart, lung, and kidney. *Pig1a* includes adult female tissues of two sample types. *Pig2b* includes tissue samples of brain, liver, pituitary, intestine, and others. *Pig4e* provides embryonic tissue RNA. Notably missing were head sensory organs, one result being that some eye, ear, nose and taste receptor genes are under-represented or fragmented in this reconstruction.

Selection of these four projects was done with the objective of collecting all transcribed genes of the pig, within constraints on effort, using the author's experience in reading RNA database information on samples. A full range of tissues, sex, development stages of samples and accuracy of sequencing instruments are important criteria for transcript reconstruction. Collecting public RNA samples that include all expressible genes requires some trial and error, or very large computational effort for well-studied organisms. Of the many RNA source projects for pig, most samples are for specific tissues, often for mutant strains, with limited sample documentation, and for less accurate sequencer instruments.

EvidentialGene methods use several gene modeling and assembly components, annotates their results with evidence, then classifies and reduces this over-assembly to a set of loci that best recovers the gene evidence. Each component method has qualities that others lack, and produces models with better gene evidence recovery. Gene reconstruction steps are (*Gilbert, 2012*): 1. produce several predictions and transcript assembly sets with quality models; 2. annotate models with available gene evidence (transcript introns, exons, protein homology, transposon, and other); 3. score models with weighted sum of evidence; 4. remove models below minimum evidence score; 5. select from overlapped models at each locus the highest score, and alternate isoforms, including fusion metrics (longest is not always best); 6. evaluate resulting best gene set (i.e., compare to other sets, examine unrecovered gene evidence); 7. re-iterate the above steps with alternate scoring to refine. Evidence criteria for genes are, in part, protein homology, coding/non-coding ratio, RNA read coverage, RNA intron recovery, and transcript assembly equivalence.

For RNA-only assembly, this paradigm is refined at step 2–4 to introduce a coding-sequence classifier (*Gilbert, 2013*), which reduces large over-assembly sets (e.g., 10 million models of 100,000 biological transcripts) efficiently, using only the self-referential evidence of coding sequence metrics (protein length and completeness, UTR excess).

CDS overlap by self-alignment identifies putative gene loci and their alternate transcripts, similarly to how CDS overlap by alignment to chromosomal DNA is used in traditional genome-gene modeling to classify loci. This CDS classifier, in tr2aacds.pl pipeline script, uses the observed high correlation between protein completeness and homology completeness, making a computationally efficient classifier that will reduce the large over-assembly set to one small enough that the additional evidence classifications are feasible to refine this rough gene set to a finished one, using evidence of protein homology, expression validity, chromosomal alignment, and others.

An automated pipeline, SRA2Genes, with methods outlined above, is used for this pig gene reconstruction. It includes RNA-seq data fetching from NCBI SRA, over-assembly of these data by several methods and parameters, transcript assembly reduction with coding-sequence classifier, protein homology measurement, sequencing vector and contamination screening, gene annotation to publication quality sequences, and preparation for submission to transcript shotgun assembly archive (TSA). Each project RNA data table is returned by SRA web query, as srapig-PRJNA416432.csv for *pig1a*.

The program is run with this data table, as for *pig1a*, to fetch data and produce cluster batch scripts for compute steps: "$evigene/scripts/evgpipe_sra2genes.pl -SRAtable srapigPRJNA416432.csv -runname=pig1a -NCPU=8 -log." Each of the four projects' RNA set was assembled this way, to the step of non-redundant gene set with alternate isoforms. Then a single gene set is produced, by combining these four reduced assemblies as input transcripts to SRA2Genes. Supplemental Archive 4 contains scripts used for this. Details of the components, their options and configurations for gene reconstruction that are used in this automated pipeline are contained in the public release of software and data (see 'Results'), as built during development with many animal and plant gene sets.

Assemblers used for all four RNA samples are Velvet/Oases (*Schulz et al., 2012*), idba_tran (*Peng et al., 2013*), SOAPDenovoTrans (*Xie et al., 2013*), and Trinity (*Grabherr et al., 2011*). K-mer sizes are computed to span the read sizes in, usually, 10 steps. As observed in this and other RNA assembly studies, k-mer of 1/2 read size produces the single most complete set, however, most k-mer sizes produce some better gene assemblies, due to wide variation in expression levels and other factors (*Zhao et al., 2011*; *Peng et al., 2013*; *Bankevich et al., 2012*). Strongly expressed, long genes tend to assemble well with large k-mer. Both non-normalized and digitally normalized (*Crusoe et al., 2015*) RNA sets were used; each method produces a somewhat different set of accurate genes.

To create a single species gene set, secondary runs of SRA2Genes are performed, starting from the combined transcript set of assembly/reductions on four samples. Several intermediate runs were used in this way, assessing gene set completeness. For instance, *pig4e* sample was added after the merging of *pig1a, 2b, 3c* samples (all adult tissues) found a deficit for embryonic genes. After merging four project over-assemblies, reference protein assessment identified fragment models. These were targeted for further assembly with rnaSPAdes (*Bankevich et al., 2012*) to finish fragment assemblies. Prior work with several methods of assembly extension has proven to be unreliable, including assemblers Oases, SOAP and idba. These typically extend fragments by sequence overlap alone, but rarely produce longer coding sequences, instead indel errors, and fused genes are frequent artifacts. rnaSPAdes, unlike the others, uses a graph of paired reads to extend partial transcripts, and may prove more reliable. Five merges of all four samples were done, with addition of new transcript assemblies to replace missing or fragmented models.

# RESULTS
## Data and software result public access
An open access, persistent repository of this annotated pig gene data set is at https://scholarworks.iu.edu/ with DOI 10.5967/K8DZ06G3. Transcriptome Shotgun Assembly accession is DQIR01000000 at DDBJ/EMBL/GenBank, BioProject PRJNA480168, for these annotated transcript sequences. Preliminary gene set is at http://eugenes.org/EvidentialGene/vertebrates/pig/pig18evigene/. EvidentialGene software package is available at http://eugenes.org/EvidentialGene/ and at http://sourceforge.net/projects/evidentialgene/.

**Table 1 Sus scrofa (pig) gene set numbers, summary output of SRA2Genes, version Susscr4EVm.**

39,879 gene loci, all supported by RNA-seq, most also have protein homology evidence
    39, 879 (100%) are protein coding, 0 are non-coding
All genes (100%) are assembled from RNA evidence, 0 are genome-modeled
25,383/39,879 (64%) have protein homology to other species genes.
316,491 alternate transcripts are at 25,512 (64%) loci, with 5 median, 12.4 ave, transcripts
    per locus, with 756 alts maximum, 1,079 loci have 50+ alts, 8,453 have 10+ alts,
27,473 (69%) have complete proteins, 12,406 have partial proteins, of 39,879 coding genes
37,918 (95%) are properly mapped to chromosome assembly ($\geq$80% align),
    1,144 partial-mapped coverage (10% < align <80%),
    817 are ~un-mapped genes (align < 10%),
6,746/37,918 (18%) are single-exon loci of those mapping $\geq$50% to genome,
    3,274 of these have homology to other species genes.
92,627 are culled loci, not in public gene set, but with some unique sequences.
    99 culls are multi-exon, well aligned; 87,515 are single exon, well aligned,
    1,082 are partially mapped, and 3,931 are poorly aligned to chromosomes.
    13,658 culls have protein homology, 78,969 lack it.
175,793 are culled alternate transcripts, at both public and culled loci, redundant
    in splicing patterns to public alternates, or lacking in alignment or evidence.
Gene locus IDs: Susscr4EVm000001t1 .. Susscr4Evm137575t1, Alternate transcripts have ID suffix t2 .. t100. Evm000001 is the longest protein, ID numbers are ordered by protein size, mostly. Culled transcripts are those initially classed as unique coding sequences, but re-classified as redundant, or lacking sufficient evidence, by chromosome alignment and homology evidence. These are separate from the public gene set as low quality, but are available as expressed transcripts, that may be recovered with further evidence.

The results of gene assembly for each of 4 data sources are summarized as *pig1a* 11,691,549 assemblies reduced to 595,497 non-redundant coding sequences (5%), *pig2b* 3,984,284 assemblies reduced to 404,908 (10%), *pig3c* 8,251,720 assemblies reduced to 564,523 (7%), and *pig4e*, a smaller embryo-only RNA set, of 1,955,018 assemblies to 134,156 (7%). These 4 reduced assemblies are then used in secondary runs of SRA2Genes, starting with these as input transcripts. Secondary runs were performed as noted in Methods, with reference homology assessment, to ensure all valid homologs are captured. Some fragment gene models were successfully improved by additional assembly with rnaSPAdes (16,168 or 5% of final transcripts, including 1,571 loci with best homology). Supplemental Archive 4 contains scripts generated by SRA2Genes and used to assemble, reduce, annotate and check sample *pig1a* on cluster compute system; these are also available in the above noted scholarworks.iu. edu repository.

The final gene set is summarized in Table 1 by categories of gene qualities and evidences. Only coding-sequence genes are reported here. The number of retained loci include all with measurable homology to four related vertebrate species gene sets, and a set of non-homologs, but expressed with introns in gene structure, two forms of gene evidence that provide a reliable criterion. The number with homology is similar to that of RefSeq genes for pig. The expressed, multi-exon genes add 15,000 loci, which may be biologically informative in further studies. The pig RefSeq gene set has 63,586 coding-sequence transcripts at 20,610 loci, of which 5,177 CDS at 5,056 loci have exceptions to chromosome location (indels, gaps, and RNA/DNA mismatch). Non-coding genes are not reported in this Evigene pig set as they lack strong sequence homology across species and are more difficult to validate.

**Table 2** *Sus scrofa* gene sets compared for gene evidence recovery: (A) conserved vertebrate genes in pig gene sets (BUSCO), (B) human reference genes (Homo sapiens RefSeq).

| Geneset | Align | Miss | Frag | Best |
|---|---|---|---|---|
| A. Vertebrate conserved genes | | | | |
| Evigene | 447 aa | 8 | 10 | 776 |
| NCBI | 440 aa | 17 | 2 | 80 |
| Ensembl | 431 aa | 14 | 20 | na |
| B. Human reference genes | | | | |
| Evigene | 97% | 0.7% | 1.4% | 30% |
| NCBI | 96% | 0.7% | 0.7% | 7% |
| Ensembl | 95% | 0.9% | 1.1% | 3% |

Note:
Scores are the count for (A), and percent of reference count ($n = 37,883$) for (B). Align = alignment to reference proteins, as percent (B) or amino average (A), Frag = fragment alignment, size <50% of reference, Miss = no alignment, Best = percent (B), or count (A) of greater alignments in pairwise match to each reference gene.

Table 1 is a computed summary of gene categories produced at the final step of SRA2Genes, following annotations and validations. The extended gene set includes culled transcript sequences, which do not meet criteria for homology or unique expression, but which pass other criteria for unique transcripts: 92,627 culled loci, and 175,793 culled alternate transcripts. Further evidence may indicate some of these are valid. The published gene data set includes mRNA, coding, and protein sequences in FastA format for the public set (pig18evigene_m4wf.public mrna, cds and aa), and the culled set (pig18evigene_m4wf.xcull_ mrna, cds and aa). There are two sequence object-annotation tables, pig18evigene_m4wf.pubids (gene locus and alternate public ids, object ids, class, protein and homology attributes), and pig18evigene_m4wf.mainalt.tab (locus main/alternate linkage for original object ids). A gene annotation table pig18evigene_m4wf.ann.txt contains public ids, name, protein, homology, database cross references, and chromosome location annotations. Chromosome assembly locations to RefSeq pig genome are given in pig18evigene_m4wf.mrna.gmap.gff in GFF version 3 format.

The Table 2A scores are measured against vertebrate conserved genes (BUSCO subset of OrthoDB v9, *Simao et al., 2015*). These scores are counts relative to 2,586 total conserved genes, but for the Align average of amino bases. Full is the count of pig genes completely aligned to conserved proteins. Table 2B has scores for human gene alignments, percentages relative to all reference genes found in either pig set ($n = 37,883$), calculated from table of "blastp -query human.proteins -db two_pigsets.proteins -evalue 1e-5." These proteins include 19,122 of 20,191 (95%) of human gene loci. Tables S1 and S2 have the pair-wise pig gene alignment scores from which summary Tables 2A and 2B are computed.

Average homology scores are nearly same for these gene sets (human alignment averages are Evigene 585aa, NCBI 586aa, Ensembl 568aa, Ensembl is significantly lower by *t*-test), but they differ at many individual loci. The "Best" columns in Table 2 indicate a subset of Evigene that can usefully improve the NCBI gene set: 3,200 proteins have improved human gene homology to greater or lesser extent, while 4,500 of Ensembl genes

can be improved by this metric. A total of 283 of Evigene improvements have no pig RefSeq equivalent, including the nine vertebrate conserved BUSCO genes missing from the NCBI set. A total of 121 of the improved coding genes are modeled as non-coding in RefSeq, which can be better modeled as coding genes with exceptions in chromosome mapping. A total of 548 have a RefSeq mRNA that is co-located with an Evigene model, but notably deficient in human gene alignment (i.e., a fragment or divergent model), while a majority of 1,048 improvements have small, exon-sized differences, as alternate transcripts to existing RefSeq loci.

Supplemental Information 3 contains genome map figure examples of ten Evigene improved loci, vs NCBI and Ensembl locus models. Most of these cases have gene models from all three sets, however, Ensembl is missing a model of the longest animal gene TITIN (Fig. S5) and early endosome antigen 1 (Fig. S2, EEA1). In some of these examples, NCBI and/or Ensembl have a fragment model missing more than 50% of coding exons. A chromosome assembly error is inferred at a gamma-tubulin locus, where both NCBI and Ensembl models lose coding sequence and orthology at the same map position (Fig. S3, TUBGCP6). Many structure differences are in the nature of alternate isoforms, where an Evigene isoform has greater homology to a human gene. There are many more alternate isoforms in Evigene models than in the other two gene sets.

Alternate transcripts share one or more coding exons for each gene locus, and have unique splice patterns, found with alignment to the pig chromosomes. This is a usual validation measure for alternate transcripts, and as indicated above the validation steps removed (or culled) those with redundant splice patterns. The 64% of loci with alternates (Table 1) compares to 75% of human loci with alternates, and surpasses NCBI and Ensembl pig gene sets, both with alternates at about 50% of loci. These alternate transcripts, all from RNA assemblies, contribute many unique additions that are homologous to human isoforms. Of the 19,122 human genes with pig gene homology, 14,938 have human alternates. Considering only the longest, primary human isoform, there is no significant difference in average alignment of NCBI and Evigene pig proteins, and as noted above Ensembl set has a sig. lower alignment. But when unique pig isoform alignments to human alternates are measured, the Evigene set significantly surpasses both NCBI and Ensembl alternate homology. This is the unique alternate alignment relation: pig_g1a2 × human_g1a2 is greater than pig_g1a1 × human_g1a2, where pig_g1a1 × human_g1a1 is longest alignment, and pig, human g1a2 are alternate isoforms. Evigene alternates with unique human isoform alignment contribute an added 59 aa alignment for the average human locus, 10% of primary alignment, vs 44 aa for NCBI, and 41 aa for Ensembl (significant at $p < 0.0001$), and 50% of human gene loci have one or more unique Evigene alternates with homology, vs 39% for NCBI or Ensembl alternate isoforms. These alternate isoforms are included in Table S2 homology comparisons, and gene map figures in Supplemental Information 3 show examples with Evigene alternates with improved homology.

Many of the 15,000 putative genes that lack homology to human, cow, mouse, or fish RefSeq genes do have homology by other measures. With non-redundant NCBI protein database, 11% of these have a significant match, to uncharacterized genes in other

**Table 3 Assembler method effects on Human reference gene recovery in Pig gene sets: (A) sample Pig1a (PRJNA416432) and (B) sample Pig2b (PRJNA353772).**

| Method | Miss (%) | Frag (%) | Short (%) |
|---|---|---|---|
| A. Sample Pig1a | | | |
| Velvet | 5 | 7 | 23 |
| Idba | 8 | 12 | 30 |
| Soap | 12 | 16 | 36 |
| Trinity | 20 | 28 | 49 |
| B. Sample Pig2b | | | |
| Illumina_all | 4 | 6 | 20 |
| Illum_velvet | 5 | 7 | 23 |
| PacBio+ | 12 | 15 | 33 |

Note:
Scores are percent of reference count ($n$ = 37,883) for Miss = no alignment, Frag = fragment alignment, size <50% of reference, Short = percent with size <95% of reference.

mammals or vertebrates, or endonuclease/reverse transcriptase transposon-like proteins, or as fragment alignments to characterized proteins. Coding alignment of these putative genes to the cow (*Bos taurus*) chromosome set, and calculation of synonymous/non-synonymous substitutions (Ka/Ks), identifies from 13% to 28% have coding sequence conservation, the majority not identified as having protein homology in the other tests. These putative genes may include recently duplicated and modified coding genes, ambiguous non-coding/coding genes, as well as fragments of other genes, putative transposon residue, and untranslated but expressed genome regions.

The Table 3A scores are for alignments to human gene with blastp, subset by assembler method for *pig1a* sample. Table 3B scores for the *pig2b* sample are also subset by methods for alignment to human genes. This second project collected both Illumina RNA-seq (75 bp paired reads) and PacBio (<1–2, 2–3, 3–5 kb, >5 kb single reads from Pacific Biosciences instrument) from the same set of tissue samples. This PacBio assembly, which includes improvement using the Illumina RNA with Proovread, was done by that project's authors and published in SRA, under Bioproject PRJNA351265. It is not used in this Evigene reconstruction, which uses instead Illumina sequences from the same pig sample and authors. The Ensembl pig gene build cited in comparison used these PacBio sequences only of *pig2b*, as well as the same *pig3c* RNA-seq Illumina sequences. The NCBI pig gene reconstruction likely used all or most of the same RNA samples used here.

The major option used for these various assemblies is k-mer size, the sub-sequence length for placing reads in the assembly graph structure. Different genes are best assembled with different k-mer sizes, depending on expression level, gene complexity, and other factors, that indicates why many assemblies of the same data but different options result in a larger set of accurate gene reconstructions. For Table 3A sample, with read size of 150 bp, k-mers from 25 to 125 were used. k-mer of 105 returned the most accurate genes, for both velvet and idba methods. The range k70..k125 produced 5/10 of best models, range k40..k65 produced 4/10, and range k25..k35 the remaining 1/10 of best models.

The popular Trinity method underperforms all others, due part to its limited low k-mer option.

Sample *pig2b* (Table 3B) demonstrates the value of assembling accurate gene pieces (Illumina, 80% of reads have highest quality score in SRA), over inaccurate but longer sequences (PacBio, 15% of reads have highest quality score in SRA). This project sequenced pig RNA with both technologies, and PacBio assembly software plus Illumina RNA to improve PacBio sequence quality, to produce a gene set, that is, less accurate than that produced from the Illumina-only RNA, assembled with a competent short-read assembler.

## DISCUSSION

The main result of this demonstration compared with the NCBI RefSeq pig gene set is, on average, they are equally valid by homology measures, but differ at many gene loci, with Evigene adding many alternate transcripts. The Evigene set also retains more putative loci, lacking measured homology but with other evidence, that further study will clarify their value. Improvements to the pig gene set are numerous enough to warrant updating RefSeq with those from this work. These include 1,500 missing or poorly modeled genes with homology to human, and improved vertebrate conserved genes. Between RefSeq and Evigene sets, all highly conserved vertebrate genes of the BUSCO set exist in pig. Another 3,000 improvements are mostly alternate transcripts with greater alignment to other species, by changes in an exon or two.

This Evigene set has demonstrated objectively accurate gene assemblies that improve the reference gene set of the pig model organism. It has been submitted for that purpose to NCBI as a third party annotation/assembly (TPA) of a TSA, which are International Nucleotide Sequence Database Collaboration (INSDC) classifications. There are policy reasons to limit inferential or computational TPA entries, and there are also policy reasons to accept these. On one hand, objectively accurate gene and chromosome assemblies of experimental RNA and DNA fragments are the desired contents of public sequence databases. On the other hand, having many assemblies of the same RNA or DNA fragments is confusing and could overwhelm databases devoted to experimentally derived genome sequences. This pig gene set adheres to the described policy of TPA in that (a) it is assembled from primary data already represented in the INSDC databases (SRA sequence read section); (b) it is indirectly experimentally supported by reference gene homology measures; (c) it is published in a peer-reviewed scientific journal. Additionally this gene set provides thousands of improvements to the reference gene set. The author produced no wet-lab experimental evidence, but has assembled gene sequence evidence from several sources into a gene set that substantially improves upon NCBI EGAP and Ensembl gene sets. Review of this data set, by NCBI and independent peers, weighs the above dilemma: improve public genome sequences or limit independent computational assemblies.

Animal gene set reconstructions by Ensembl and NCBI RefSeq are widely used and considered high quality, though recent published comparisons of these two are uncommon (for the special case of human genes, see *Zhao & Zhang, 2015*). This author often

**Table 4 Conserved genes in model animals, mis-modeled by three methods.**

| Gene set | Pig | Cow | Mouse | Rat | Fish | Human |
|----------|-----|-----|-------|-----|------|-------|
| Evigene  | 18  | –   | –     | –   | 14   | –     |
| NCBI     | 19  | 22  | 9     | 24  | 25   | 1     |
| Ensembl  | 34  | 58  | 5     | 32  | 79   | 1     |

**Notes:**
Mis-model is Missing + Fragmented, calculated for 2,586 vertebrate conserved genes, as per Table 2A. Gene sets of year 2018 from NCBI, Ensembl, and two of Evigene are as listed in "Data and Software Citations."

compares NCBI and Ensembl genes when constructing Evigene models, observing that NCBI methods now commonly surpass those of Ensembl, as is found in these pig gene sets. Table 4 indicates this for reconstruction of conserved genes in five model animals, all with errors above the human set that are correctable (e.g., for pig NCBI+Evigene). The NCBI improvement is likely due in part to NCBI EGAP's more extensive use of RNA-seq and transcript assemblies, and use of transcript evidence-based exceptions to a rule that gene models must align fully to chromosome assemblies. Neither NCBI EGAP nor Ensembl produce de novo assemblies of RNA-seq. These projects, however, can and do use assembled transcripts from INSDC public databases. Evigene's de novo assembled genes can thus improve these other widely used gene sets.

Combining and selecting by evidence criteria the assemblies of several methods improves gene reconstruction to a higher level of accuracy. The individual methods return from 77% (Velvet) down to 50% (Trinity) of the best gene models, and a hybrid PacBio +Illumina assembly is intermediate at 66%. K-mer sizes are an important parameter, as noted by others: "smaller values of k collapse more repeats together, making the assembly graph more tangled. Larger values of k may fail to detect overlaps between reads, particularly in low coverage regions, making the graph more fragmented" (SPAdes, *Bankevich et al., 2012*). Alternate isoforms of each gene, which share exons and differ in expression levels, are more accurately distinguished from other genes at large k-mer sizes (idba_tran, *Peng et al., 2013*). These results are consistent with multi-method reconstructions for arabidopsis, corn, zebrafish, mosquitos, and water fleas.

The main flaw in this Evigene pig set is incomplete reconstruction of many genes, especially longer ones. While this is not always a problem with RNA-only assemblies, it is a common one. Importantly, there does not appear to be a reliable method for improving gene assemblies identified as fragmentary, using de novo RNA assembly. While there are several methods that attempt to address this, those tested by the author are unreliable. A trial of rnaSPAdes to extend fragments did improve some genes, but not as many as the RNA data warrants.

A second flaw in EvidentialGene's method of classifying loci from self-referential alignment of coding sequences is that some paralogs are confused as alternate transcripts of the same locus. With high sequence identity, paralogs align to each other similarly to transcripts of one locus (a class termed "altpar" or "paralt"), though with mismatches that chromosome alignment can resolve. This has been measured at a rate of about 5% for reference gene sets of mouse and zebrafish, and 3% for arabidopsis; a smaller 0.5% portion of alternates at one locus are misclassified as paralog loci. Several de novo gene

assembly methods that classify loci have similar altpar confusion, as RNA-seq reads are often shared among paralogs as well as alternate transcripts. These altpar transcripts have not been resolved for this pig gene set, though it is an improvement in development.

This demonstration excluded the use of chromosomes and other species genes to assemble or extend assemblies. Both methods can be employed to advantage to reconstruct genes, where there are few errors in these additional evidences. An important reason to limit initial gene reconstruction to RNA-only assembly is to avoid compounding errors from several sources. This limited-palette reconstruction is validated with independent evidence from genomic DNA and other species sources; genes identified as mis-assembled, or missing, in such RNA-only sets can be improved with these other methods. Many discrepancies between RNA-only reconstruction and the other evidences are from flaws in chromosome assemblies or other species genes that can be identified with careful evaluations.

Gene transcripts from any source, such as EST and PacBio, may be added into SRA2Genes pipeline. Excluded from this reconstruction are the extensive public set of pig ESTs, and the PacBio+Illumina assembly from the same study as *pig2b*. These contribute a small number of improved transcripts not in this EvidentialGene set (8 missed human orthologs in ESTs, 12 EST, and 24 PacBio with significant improvements), and are used in the RefSeq set. However, as these are already in the public databases, this demonstration reconstruction adds no value to them.

While these gene data and paper were in review at repositories, *Zhao et al. (2018)* pre-published a reconstruction of pig genes, with newly sampled proteomic and transcriptomic sequences. The authors provide public access to these under BioProject PRJNA392949 for SRA RNA-Seq, and a bioRxiv preprint with sequences of 3,703 novel protein isoforms. The experimental design of this work is well suited to gene set reconstruction, as it sampled 34 tissues of adult male, female and juvenile pigs. Unlike the samples winnowed from prior SRA entries by this author, each from a pig portion, this new work is comprehensive in collecting expressed and translated genes.

This pre-published gene set is compared to the same RefSeq gene set and chromosome assembly as this paper. In brief, of the 3,700 novel proteins, most align to other gene sets and chromosome assembly: 74% are contained in this paper's transcripts, 65% are contained in the RefSeq transcript set, and 61% are contained in the pig chromosome set, at 75% or greater alignment (protein to RNA/DNA aligned with tBLASTn). Nonetheless, most of these novel proteins do not have a protein equivalent in the gene sets: about 800 novel proteins align to Evigene proteins, and about 600 to RefSeq proteins. A main difference here lies in measures from RNA to protein, including new alternate transcripts and discrepancies in RNA to protein reconstruction, rather than in newly identified gene loci, and is beyond scope of this note to resolve. A rough draft with SRA2Genes of this recent RNA-Seq, assembling only well-expressed genes, contains about the same 74% of novel proteins as for this paper's set. An application suited to SRA2Genes is to update with these completely sampled pig genes, including depositing an improved version to Transcriptome Shotgun Assembly public database for further uses.

## CONCLUSIONS

The SRA2Genes pipeline is demonstrated, for the pig model organism, as a reliable gene reconstruction method, useful to other projects and for improving public reference gene sets. The resulting complete transcriptome assembly of pig fills a void at public repositories. Reconstruction from RNA only provides independent gene evidence, free of errors and biases from chromosome assemblies and other species gene sets. Not only are the easy, well known ortholog genes reconstructed well, but harder gene problems of alternate transcripts, paralogs, and complex structured genes are usually more complete with EvidentialGene methods.

## DATA AND SOFTWARE CITATIONS

NCBI pig gene set used in comparison, from ftp://ftp.ncbi.nlm.nih.gov/refseq/S_scrofa/mRNA_Prot/pig.1.rna.gbff.gz, accessed on 27 Apr 2018.

Ensembl pig gene set used in comparison, from ftp://ftp.ensembl.org/pub/release-93/sus_scrofa/pep/Sus_scrofa.Sscrofa11.1.pep.all.fa.gz, accessed on 28 Jul 2018.

NCBI RefSeq pig chromosome assembly Sscrofa11.1, accession: GCF_000003025.6, dated 2017-2-7, is used for chromosome mapping.

NCBI RefSeq gene sets used as reference genes are H_sapiens, M_musculus, B_taurus, and D_rerio, accessed at same location and date as pig genes. Ensembl gene sets for comparison of same species and date are from ftp://ftp.ensembl.org/pub/release-93/{species}/pep/.

Evigene zebrafish (D_rerio) gene set is at http://hdl.handle.net/2022/22652 (IUScholarWorks)

RNA data sources with NCBI BioProject ID are

SRA data pig1a: PRJNA416432 (China Agricultural University),

SRA data pig2b: PRJNA353772 (Iowa State University, USDA-ARS),

SRA data pig3c: PRJEB8784 (Univ. Illinois),

SRA data pig4e: PRJNA255281 (Jiangxi Agricultural University, Nanchang, China).

RNA data query for the pig used to select these four, at http://www.ncbi.nlm.nih.gov/sra query=(("biomol transcript"[Properties]) AND "platform illumina"[Properties]) AND "library layout paired"[Properties] AND "Sus scrofa"[Organism]

RNA long-read SRA query for the comparison data set of Table 3B: query=((("biomol transcript"[Properties]) AND "platform pacbio smrt"[Properties])) AND "Sus scrofa"[Organism]

The SRA read table of these data sets is the starting point for SRA2Genes, and provided at http://eugenes.org/EvidentialGene/vertebrates/pig/pig18evigene/

Expressed sequences of the pig from dbEST, by Sanger and 454 sequencing (max length 900 bases), from projects reported in PubMedID:14681463, dbEST n = 304,418, and PubMedID: 17407547, dbEST n = 716,260.

Vertebrate conserved single-copy genes, of OrthoDB v9 (http://www.orthodb.org), BUSCO.py software, with hmmer (v3.1, http://hmmer.org/) (Waterhouse et al. 2013, Simao et al., 2015).

Software components of EvidentialGene SRA2Genes:

fastq-dump, of sratoolkit281, https://www.ncbi.nlm.nih.gov/sra/docs/toolkitsoft/

blastn, blastp of https://blast.ncbi.nlm.nih.gov/ (*Altschul et al., 1990*)

vecscreen at http://ncbi.nlm.nih.gov/tools/vecscreen/ and tbl2asn at http://ncbi.nlm.nih.gov/genbank/tbl2asn2/

fastanrdb, of exonerate, https://www.ebi.ac.uk/about/vertebrate-genomics/software/exonerate (*Slater & Birney, 2005*)

cd-hit, cd-hit-est, of https://github.com/weizhongli/cdhit/ (*Li & Godzik, 2006*)

normalize-by-median.py, of khmer, https://github.com/ged-lab/khmer (*Crusoe et al., 2015*)

velvet, oases of velvet1210 assembler, https://www.ebi.ac.uk/~zerbino/oases/ (*Schulz et al., 2012*)

idba_tran, of idba assembler, https://code.google.com/archive/p/hku-idba/downloads/ (*Peng et al., 2013*)

SOAPdenovo-Trans, http://soap.genomics.org.cn/SOAPdenovo-Trans.html (*Xie et al., 2013*)

Trinity, of trinityrnaseq assembler, https://github.com/trinityrnaseq/trinityrnaseq (*Grabherr et al., 2011*)

rnaSPAdes, of SPAdes assembler, http://cab.spbu.ru/software/spades/ (*Bankevich et al., 2012*)

International Nucleotide Sequence Database Collaboration policy documents pertaining to these data:

About TSA, https://www.ncbi.nlm.nih.gov/genbank/TSA

About TPA, https://www.ncbi.nlm.nih.gov/genbank/TPA

TPA FAQ, https://www.ncbi.nlm.nih.gov/genbank/tpafaq

TPA-Inferential, https://www.ncbi.nlm.nih.gov/genbank/TPA-Inf.

## ACKNOWLEDGEMENTS

XSEDE/TeraGrid shared computational resources, for a decade of development and implementation, Award# MCB100147, to Genome Informatics for Animals and Plants, D.G. Gilbert. IUScholarWorks staff, including Richard Higgins, for providing a permanent open-access repository of EvidentialGene animal and plant gene sets. NCBI GenBank submissions staff is thanked for reviewing effort to deposit TPA/TSA gene data sets.

### Funding

The author received no funding for this work.

### Competing Interests

The authors declare that they have no competing interests.

## Author Contributions

- Donald G. Gilbert conceived and designed the experiments, performed the experiments, analyzed the data, contributed reagents/materials/analysis tools, prepared figures and/or tables, authored or reviewed drafts of the paper, approved the final draft, all other.

## Data Availability

An open access, persistent repository of this annotated pig gene data set is at IUScholarWorks: DOI 10.5967/K8DZ06G3.

The Transcriptome Shotgun Assembly accession is DQIR01000000 at DDBJ/EMBL/GenBank, BioProject PRJNA480168, for these annotated transcript sequences;

Data and software are also at https://sourceforge.net/projects/evidentialgene/files/evigene_pig_susscr2018/ and http://eugenes.org/EvidentialGene/vertebrates/pig/pig18evigene/.

## Supplemental Information

Supplemental information for this article can be found online at http://dx.doi.org/10.7717/peerj.6374#supplemental-information.

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
