# Peer review of "Genes of the pig, *Sus scrofa*, reconstructed with EvidentialGene"

_PeerJ, doi:10.7717/peerj.6374_

## Round 0.1 · original submission · Major Revisions

Please address all the critical issues raised by the reviewers and revise your manuscript accordingly.

Reviewer 1 ·

Basic reporting

Basic reporting is generally sound, however there are several sections that could do with proof checking to provide a clearer understanding of what the author means, please see general comments below. I also think some time spent checking grammar and spelling would be warranted.

One area that could do with improvement is in regards to context/aims in the introduction. E.g. line 44 please give the reasons/context for why gene set reconstruction fails. Based on this, set out how the methods applied here, inc the new automated pipeline, aims to address this. – also you don’t say what it is you are demonstrating in the para (which is important)

While I commend the author for providing a thorough data and software citations section that provides links to all the projects/data and software used, I do feel that it does a disservice to their authors/contributors by not citing the appropriate literature, which should be rectified in the revision.

Experimental design

The method section, while presenting an ok overview of the board steps involved does lack sufficient detail to be replicable. The whole section reads rather jumbled and does not flow logically. I would like to see clear description of steps involved and citations to techniques implemented if not described here.

Line 70 – what types of modelling, what are their qualities over one another – cite works that has already shown this if available.
99 – what assembly methods and parameters – while these are listed later (128) parameter choice is not given. Maybe a summary table of assembler and parameters would help?
104-106 you say data was selected on tissue sample, methodology and others; then point to data section – yet these details are not given in that section. Please be clear about how you selected data. i.e to cover the majority of tissue types, only selecting for Illumina PE data etc etc
130 – k-mer sizes selected would also be good to should in the table suggested above.
134 – provide detail on normalization methods.

Validity of the findings

The implemented pipeline/method shows substantial improvement in terms of gene set reconstruction and will be useful to future work, which is in line with the stated conclusions. The data seems robust and sound (I like the persistent DOI’s that provide clear data and results).

However, I find the results section particularly hard to follow and glean the important information from. It reads largely as text pointing to where results data can be found (e.g.175-186 and 192-201, 248 onwards) rather than actually presenting results in a clear and interpretable manner.

161 How many subsequent runs – how much did they iteratively improve the results?
228-229 is more of a description of a dataset rather than results of its analysis.
248-253 – where are the results in this section?

I feel the discussion is missing important evidence (in terms of citations or data shown in results) to back up statements made. E.g. Lines: 301-303, 311-314). An important consideration is that the authors ‘sells’ the note as a demonstration of his pipeline. While examples of how to execute it are available online I think a more clear demonstration in this paper is warranted. Be this either as a clear step by step example with embedded code (which would be highly valuable to readers) or a clear flow diagram (and associated text) that clearly shows the computational steps of the pipeline.

Additional comments

It is unfortunate that this note reads as being largely unfinished, as I am keen to see it published because the method is exciting and valuable. I think it is easily rectifiable with further work.

Please give full species names in para 4 of Intro.
The language used can be difficult to follow, for example the following sentences could do with rewording to improve their clarity. Some examples follow, but it is not an exhaustive list: Lines: 38, 97, 106, 111, 130-135,160, 163, 190, 205, 268-271
111-115 does not fit in methods – maybe this should be moved into the discussion in a considerations section or removed entirely.

Reviewer 2 ·

Basic reporting

In this manuscript, the author described the result of his annotation pipeline “the EvidentialGene” for a pig genome. However, I could not evaluate it properly due to the lack of critical information in the current manuscript. He cited his previous works published as conference posters, which does not have a full description of his method.

For example, gene construction part is critical in this process, but he only described all steps in one paragraph (2nd paragraph of the method), and his cited poster also does not have a full explanation: how to make a score for each step? What is the minimum evidence score? What kind of evidence was used, and how were they weighted? I think this manuscript would be better to describe the Evidential Gene pipeline more technically, with a pig transcriptome as an example.

There is no figure in this manuscript, but all tables need to be revised substantially. Table 1 is hard to read the information. Table 2 needs more information to understand what “Full”, “Align”, Miss”, and “Best” mean. Table 3 also requires more explanation about the criteria for “Miss”, “Frag”, and “Short”. Also, the author mentioned four data in the manuscript, but he only reported two cases here. The more comprehensive report should be presented for each table.

Experimental design

De novo transcriptome assembly is one of the key steps in this method, but the detailed experimental procedure is lacking. For each step, the author needs to provide detailed parameters, and how to evaluate a tremendous amount of transcriptome candidates for refinement. In the current manuscript, the author only reported the number of outputs, without explaining how that outcome was produced. The author also mentioned multiple assemblers, but it is not clear which assembler is used for each data. It should be clarified.

More explanation is required why a pig is a good case to apply the EvidentialGene pipeline. Although the author described some reasons in Introduction (many data, good quality of genome, etc), the genome is published already and major genome resources such as EnsEMBL and UCSC provided comprehensive annotation, and a recent version of EnsEMBL (Sscrofa11.1) also incorporated PacBio long reads and RNA-seq data for their annotation. At least, the author needs to compare current annotation with the result of EvidentialGene, which is completely missing in this manuscript.

Validity of the findings

I agree with the author that the approach presented in this manuscript (EvidentialGene) is useful to evaluate different version of genome annotation in many organisms. However, current manuscripts need to be revised substantially, to support its contribution. Without systematic comparison with current annotation, it is hard to know the benefit of this approach. Although the author provides the comparison with RefSeq data, at least two other major annotation data (EnsEMBL, UCSC) should be analyzed.

The conclusion is also not well presented by data. The main description is the number of transcripts and their associated loci, but the more important thing is how to reduce false positive (mis-annotated genes), or false negative (missing genes). By analyzing genes not identified in this pipeline, and those only identified here, the author may argue the benefit of the EvidentialGene to reduce these errors.

Reviewer 3 ·

Basic reporting

This manuscript was well-written and relatively easy to understand. One point that I think really could be improved (for clarity) is how the author refers to the different data sets used. I think I counted 4 different methods used (full database name, submitter, pig1a or just pig 1) throughout the manuscript. Please select one method. My preference would be something easy for the reader to relate to; ie. submitting institution, or reason the data set was chose, rather than a 5 letter code followed by 4-6 integers or a random number assignment. My only other item to mention is Table 1 looked more like a slide than a table. Otherwise this was a very easy manuscript to read.

Experimental design

I think the overall design of this study was quite sound. However, the description of some methods are clearly lacking. First and foremost, the gene sets were mapped to the swine genome, but the author never states what build of the swine genome was used. This is a must! Also, I hope it was build 11.1, because if it wasn't, then the author needs to go back and re-map these loci.
Second, the author selected 4 public data sets. The description of these data sets (lines 117-126) is extremely inadequate! I suspect each data set was selected for a reason. I suspect the first data set was selected as it probably has the most reads, the last data set as it had low read depth and may contain more developmental genes and the second data set may have been selected as it was the only one with Iso-Seq data (PacBio longread), but this isn't specifically stated. Also, lines 120-121 states all data were short-read, paired end data, but the results indicate PacBio reads were used (lines 231-236) for the second data set, this needs to be corrected.

Validity of the findings

The results of the new analysis pipeline appear to fix many issues with current automated annotation pipelines. While there are a number of different methods that can be used to annotated transcripts, no single method is perfect. Thus, it is quite useful to have a variety of methods developed. Comparison among results of different methods can either validate annotation or point researchers to regions that may need additional experimentation to resolve.
If I understand correctly, the final list of transcript are for protein coding mRNA. How are non-coding transcripts identified and annotated with the current methods? Do some (or all) end up in the culled transcripts group? If they do, I think it may be interesting if the author could elaborate more on these important transcripts.

Additional comments

Overall I really enjoyed this manuscript and believe it offers a valuable method for semi-automated annotation of transcripts. I also thought the discussion on how results are affected by k-mer length selection was very enlightening. While I am sure this method has some limitations (as all methods do) it clearly is superior to the current methods used for the pig.

---

## Round 0.2 · Minor Revisions

Please address the remaining (minor) points raised by the reviewers.

Reviewer 1 ·

Basic reporting

no comment

Experimental design

no comment

Validity of the findings

no comment

Additional comments

Firstly, thank you for the effort in revising the manuscript – especially the inclusion of additional comparisons to NCBI and ENSEMBL sets and the much needed methodological detail (inc the script archive), which improve not only rigor but increase the papers usefulness.

The paper is much improved since my last evaluation. However, I still find the language used hard to digest, but this may be personal preference of writing styles. I would request an additional careful read through before final acceptance.

Minor comments:

Line 66. “The automated pipeline of SRA2Genes is introduced and will be detailed in other papers.”

Change to say either “The automated pipeline of SRA2Genes is introduced” or “The automated pipeline of SRA2Genes is introduced and the EviGene reconstruction method will be detailed in forthcoming papers.” I don’t like “here’s this, but im not talking about it yet” – my preference would be for the first edit.

Line 250. “There are many more alternate isoforms in Evigene models than in the other two gene sets.” Does the author believe these are real isoforms, or potential misassemblies? Does mapping the RNA-Seq data (esp pac bio data) confirm that some of these are in fact real? This would add additional evidence that evigene produces a ‘better’ gene set. Otherwise, maybe evigene is producing a ‘better’ passing set, but the alt set has much higher prevalence of false isoforms?

Line 330: Im not happy with comments such as “This author often compares NCBI and Ensembl genes when constructing Evigene models, observing that NCBI methods now commonly surpass those of Ensembl, as is tabulated in these pig gene sets. “ . Unless the author can back them up with valid data across multiple species. Either just say that NCBIs current pig gene set is better, or back up your generalisation with evidence.

Reviewer 3 ·

Basic reporting

No comment

Experimental design

No comment

Validity of the findings

No comment

Additional comments

The revised version has addressed all of my concerns. I have a few minor edits that need to be done.
Line 19, I think indicating improved alternate transcripts and more full-length transcripts would be better than just stating "other improvements".
Line 39 change 'pubic' to 'public'.
Line 74 insert 'and' in front of 'instrument'.
Line 97 delete the first 'are' on this line.
Line 196 are the 5177 exceptions coding-sequences or loci?
Line 395 a full citation for Zhao is required.
Table 2 does not define 'Full' and has a definition for 'Short' that is not used.

---

## Round 0.3 · accepted · Accept

All the remaining critical issues were successfully addressed and the manuscript was amended accordingly. Therefore, the revised version is acceptable now.

#